# Robust genetic codes enhance protein evolvability

**Hana Rozhoňová**[1,2]\*, **Carlos Martí-Gómez**[3], **David M. McCandlish**[3], **Joshua L. Payne**[1,2]\*

**1** Institute of Integrative Biology, ETH Zürich, Zürich, Switzerland, **2** Swiss Institute of Bioinformatics, Lausanne, Switzerland, **3** Simons Center for Quantitative Biology, Cold Spring Harbor Laboratory, Cold Spring Harbor, New York, United States of America

\* hana.rozhonova@env.ethz.ch (HR); joshua.payne@env.ethz.ch (JLP)

## Abstract

The standard genetic code defines the rules of translation for nearly every life form on Earth. It also determines the amino acid changes accessible via single-nucleotide mutations, thus influencing protein evolvability—the ability of mutation to bring forth adaptive variation in protein function. One of the most striking features of the standard genetic code is its robustness to mutation, yet it remains an open question whether such robustness facilitates or frustrates protein evolvability. To answer this question, we use data from massively parallel sequence-to-function assays to construct and analyze 6 empirical adaptive landscapes under hundreds of thousands of rewired genetic codes, including those of codon compression schemes relevant to protein engineering and synthetic biology. We find that robust genetic codes tend to enhance protein evolvability by rendering smooth adaptive landscapes with few peaks, which are readily accessible from throughout sequence space. However, the standard genetic code is rarely exceptional in this regard, because many alternative codes render smoother landscapes than the standard code. By constructing low-dimensional visualizations of these landscapes, which each comprise more than 16 million mRNA sequences, we show that such alternative codes radically alter the topological features of the network of high-fitness genotypes. Whereas the genetic codes that optimize evolvability depend to some extent on the detailed relationship between amino acid sequence and protein function, we also uncover general design principles for engineering nonstandard genetic codes for enhanced and diminished evolvability, which may facilitate directed protein evolution experiments and the bio-containment of synthetic organisms, respectively.

**Data Availability Statement:** All relevant data are within the paper, on GitHub (https://github.com/parizkh/rewired_codes_landscapes), and in the public repository Zenodo (10.5281/zenodo.10677993).

## Introduction

Proteins are the workhorses of the cell. They are the building blocks of cellular infrastructure, they transport molecules, regulate gene expression, and catalyze essential biochemical reactions. How do such protein functions evolve? The classic metaphor of the adaptive landscape is helpful to conceptualize this process [1]. An adaptive landscape is a mapping from genotype space onto fitness or some related quantitative phenotype, which defines the "elevation" of each coordinate in this space. For proteins, genotype space comprises the set of all possible

**Funding:** This work was funded by Swiss National Science Foundation (https://www.snf.ch; grants PP00P3_202672 and 310030_192541 to J.L.P.), NIH (https://www.nih.gov/; grant R35GM133613 to D.M.M.), an Alfred P. Sloan Research Fellowship (https://sloan.org/; to D.M.M.), and additional funding from the Simons Center for Quantitative Biology at Cold Spring Harbor Laboratory (https://www.cshl.edu/research/quantitative-biology/; to D. M.M.). The funders had no role in study design, data collection and analysis, decision to publish, or preparation of the manuscript.

**Competing interests:** The authors have declared that no competing interests exist.

amino acid sequences of a given length [2] and the quantitative phenotypes of these sequences include catalytic activity, folding stability, and binding affinity. The evolution of protein function can then be viewed as a hill-climbing process in such a landscape, in which mutation and natural selection tend to drive evolving populations toward adaptive peaks of improved functionality [3].

Central to this process is evolvability—the ability of mutation to bring forth adaptive phenotypic variation [4,5]. For short-term, one-step adaptation, evolvability depends on the immediate mutational neighborhood of a protein sequence (Fig 1A). That is, it depends on the amount of adaptive phenotypic variation accessible via point mutation. For longer-term, multi-step adaptation, evolvability depends on the topography of the adaptive landscape. A smooth single-peaked landscape facilitates evolvability, because mutation can easily bring forth adaptive phenotypic variation from anywhere in the landscape, except atop the global peak; in contrast, a rugged landscape diminishes evolvability, because its adaptive valleys often preclude the generation of adaptive phenotypic variation [4,6,7] (Fig 1B).

What determines whether a protein's adaptive landscape is smooth or rugged? One primary factor is the genetic code an organism uses for translation. Its importance arises because it determines which amino acid changes are accessible via alteration of a single nucleotide. For example, under the standard code, point mutations to the CUG codon can change the amino acid leucine to methionine (AUG), valine (GUG), proline (CCG), glutamine (CAG), and arginine (CGG), but not to any other of the remaining 14 amino acids. A genetic code thus defines which protein sequences are "near" one another in sequence space [9], and which mutational paths to adaptation are closed or open (Fig 1C).

The structure, history, and evolutionary implications of the standard genetic code have fascinated scientists for decades [10–14]. Given the nearly infinite space of alternatives, why did life converge on the standard genetic code? What makes it so special? Answers to this question are typically based on comparisons of the properties of the standard genetic code to those of hypothetical, alternative codes [15,16], of which there are many [17]. Even if one maintains the degeneracy of the standard code, but simply randomizes which amino acids are assigned to which codon blocks, there are $20! \approx 10^{18}$ possible rewirings. By sampling a large number of such rewired codes, one can ask whether a given quantitative property of the standard genetic code has a value higher or lower than expected by chance. For example, using a measure of so-called "error tolerance" based on how well point mutations preserve polar requirement (a measure of hydrophilicity), and taking into consideration mutation bias toward transitions relative to transversions, Freeland and Hurst [16] showed that only one in a million rewired codes preserves the hydrophilicity of amino acids to a greater extent than the standard genetic code. The standard genetic code is thus highly robust to this form of error, in that point mutations and mistranslations tend to cause minor changes to this physicochemical property of amino acids.

What are the implications of code robustness for protein evolvability? By definition, a robust genetic code limits the amount of phenotypic variation that point mutations can cause. However, opinions differ on whether this hinders or facilitates evolvability. Inspired by Fisher's geometric model [18], early theoretical work argues that code robustness may facilitate protein evolvability exactly because it minimizes the effects of mutations, thus increasing the probability that mutations will be adaptive [19]. Indeed, by analyzing the fitness effects of point mutations to the antibiotic resistance gene TEM-1 $\beta$-lactamase and 2 influenza hemagglutinin inhibitor genes, it has been shown that missense mutations are enriched for adaptive amino acid changes, relative to amino acid changes that require multiple point mutations [20,21]. In contrast, more recent theoretical work [22], motivated by advances in synthetic biology [23–26], argues that protein evolvability can be enhanced by reducing code robustness,

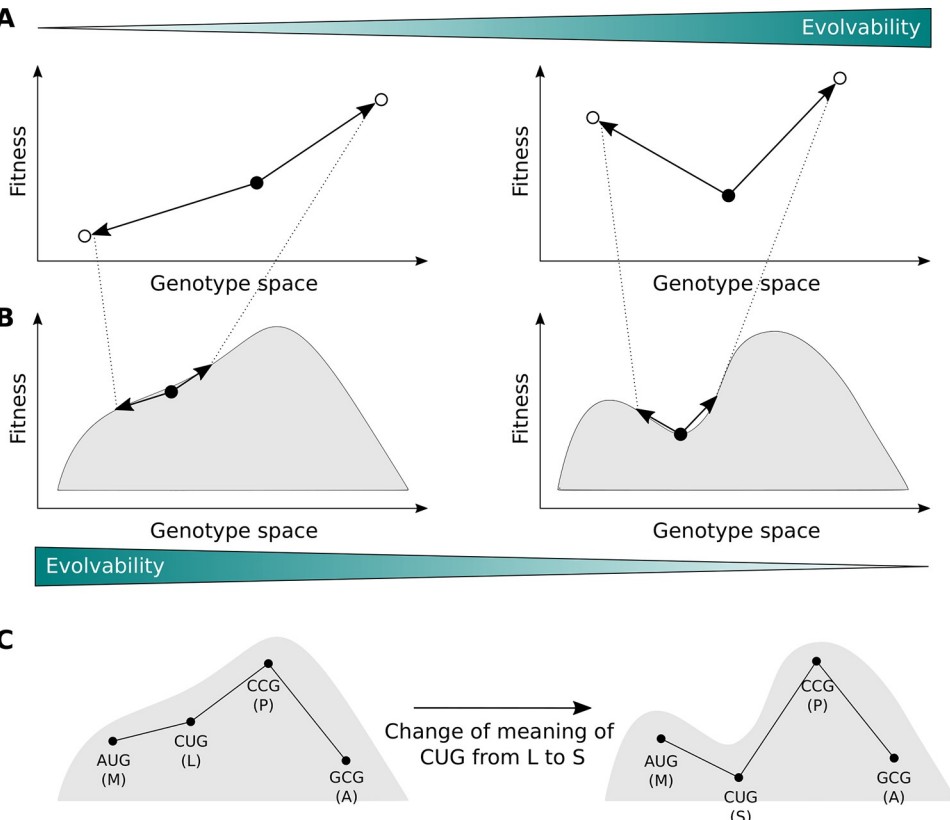

**Fig 1. Evolvability and adaptive landscapes.** (A) In one-step adaptation, evolvability depends on the amount of adaptive phenotypic variation accessible via point mutation. Therefore, the genotype shown with a filled circle in the right panel is more evolvable than the one shown in the left panel. (B) Zooming out and considering multi-step adaptation, landscape topography becomes important. Smoother landscapes promote evolvability (left panel), whereas rugged landscapes hinder evolvability (right panel), because an evolving population is more likely to be trapped on a local optimum. (C) Landscape topography is influenced by the genetic code. As a toy model, a sequence consisting of a single codon is shown. Under the standard genetic code, there is a single peak, which is also a global optimum (left panel). If the meaning of the CUG codon is changed from leucine to serine (as is the case in some yeast species [8]), an adaptive valley is formed (right panel). The population now cannot leave the local optimum consisting of the AUG codon without crossing a maladaptive valley.

because by doing so one can increase the number and diversity of amino acids accessible via point mutation to any codon.

Whether code robustness hinders or facilitates protein evolvability therefore remains an open question. Whereas steps have been taken towards answering this question [20,21,27–29], these studies suffer from at least 1 of 2 key limitations. The first is a focus on how missense mutations change the physicochemical properties of amino acids [15,16,28], rather than how missense mutations change the phenotype of a protein (e.g., its stability or catalytic activity) or the corresponding fitness of an organism. The second limitation is a lack of suitable data, with studies relying on a purely theoretical model of landscape topography [28], a categorical, rather than quantitative, phenotype [27], an incomplete fitness landscape [20], or assumptions of additivity regarding the combined effects of mutations [29]. We therefore do not know how the structure of a genetic code, standard or otherwise, influences the evolvability of proteins beyond one-step adaptation. This is an important knowledge gap, because protein evolution often proceeds via a sequence of adaptive mutations that improve protein function, as evidenced by comparisons of orthologous sequences [30,31] and directed protein evolution

experiments [32,33]. Moreover, given the increasing interest in engineering nonstandard genetic codes [24], it is desirable to deduce design principles for engineering genetic codes with reduced or enhanced evolvability, as these might be used to form a "genetic firewall" [34] or accelerate directed evolution [22], respectively.

Here, we overcome the limitations of prior studies using experimental data from massively parallel sequence-to-function assays [35]. In particular, we use combinatorially complete data, which provide a quantitative characterization of protein phenotype for all possible combinations of $20^L$ amino acid sequence variants at a small number $L$ of protein sites [36–40]. These data facilitate the construction of complete adaptive landscapes without assumptions regarding the combined effects of individual mutations (e.g., additivity). Importantly, the combinatorially complete nature of these data allows us to construct such landscapes under arbitrary genetic codes. The reason is that, no matter which code we use, we are guaranteed that each of the $4^{3L}$ possible mRNA sequences can be computationally translated into an amino acid sequence with an experimentally assayed phenotype. We characterize the topographies of 6 such empirical adaptive landscapes under the standard genetic code, as well as under hundreds of thousands of rewired codes, and perform population-genetic simulations on these landscapes. We show that robust genetic codes tend to produce smooth adaptive landscapes with few peaks and, consequently, allow evolving populations to reach on average higher fitness. Thus, the robustness of a genetic code not only helps to mitigate the potentially deleterious effects of replication and translation errors, but it also transforms the problem of molecular evolution from one that depends on the vicissitudes of individual mutations into one where evolving populations can readily find mutational paths toward adaptation.

## Results

### Data

We construct empirical adaptive landscapes using 6 combinatorially complete data sets for 4 proteins. The first protein is GB1, a Streptococcal protein that binds immunoglobulin [41,42]. Wu and colleagues [36] experimentally assayed the binding affinity of GB1 to immunoglobulin for all $20^4 = 160,000$ amino acid sequences at 4 protein sites (V39, D40, G41, and V54; Fig A in S1 Text), which are known to interact epistatically and influence binding affinity [43]. In particular, they measured the relative frequencies of sequence variants before and after selection for binding immunoglobulin. Binding affinities are then defined as log enrichment ratios (Methods).

The second protein is ParD3, a bacterial antitoxin that is part of the ParD-ParE family of toxin-antitoxin systems, which are commonly found on bacterial plasmids and chromosomes [44]. Such systems comprise a toxin that inhibits cell growth unless bound and inhibited by the cognate antitoxin. Lite and colleagues [37] experimentally assayed bacterial cell growth for all $20^3 = 8,000$ amino acid sequence variants at 3 sites in ParD3 (D61, K64, E80; Fig A in S1 Text), in the presence of its cognate toxin ParE3, as well as a related, but non-cognate toxin ParE2. This resulted in 2 data sets, 1 per toxin, in which cell growth was used as a quantitative readout of the degree to which individual ParD3 variants antagonize a given toxin.

The third protein is ParB, a DNA-binding protein crucial for bacterial chromosome segregation [45]. The binding site of ParB, *parS*, is a palindrome of GTTTCAC. Jalal and colleagues [39] experimentally measured the binding affinity of ParB to the cognate DNA sequence, *parS*, as well as a related DNA-binding site, *NBS* (palindrome of ATTTCCC), for all $20^4 = 160,000$ variants at 4 positions (R173, T179, A184, and G201; Fig A in S1 Text). This again resulted in 2 data sets, 1 per DNA-binding site.

The fourth protein is dihydrofolate reductase (DHFR), an essential metabolic enzyme in *Escherichia coli*. Papkou and colleagues [40] generated all possible $64^3 = 262{,}144$ combinations of codons at 3 positions (A26, D27, L28) of the corresponding *folA* gene. Missense mutations at these positions are known to confer resistance to the antibiotic trimethoprim [46,47]. Using a mass-selection experiment, Papkou and colleagues [40] measured the fitness of each variant in the presence of a sublethal dose of trimethoprim. The majority (89.7%) of the variants are nonfunctional, in that they are sensitive to trimetophrim.

Following the protein evolution literature [3,36,48], we assume that fitness is directly proportional to binding affinity (GB1, ParB) or growth rate (ParD3, DHFR), and will use the term "fitness" generically for all landscapes from now on. Using the raw measurements described above (binding affinities and cell growth), we inferred the fitness values, as well as imputed the missing sequence variants (6.6% of the GB1 data set) using empirical variance component regression [49] (Methods and Fig B in S1 Text).

For each of the 6 data sets, we constructed adaptive landscapes using the standard genetic code, as well as hundreds of thousands of rewired codes. Specifically, we represented each mRNA sequence of length 12 (GB1, ParB-*parS*, ParB-*NBS*) or 9 (ParD-ParE2, ParD-ParE3, and DHFR), respectively, as a vertex in a mutational network and connected vertices with an edge if their corresponding sequences differed by a single point mutation [50] (Methods). We labeled each vertex with the fitness of its corresponding translation under a given genetic code, thus defining the "elevation" of each coordinate in genotype space.

## More robust codes cause smoother adaptive landscapes

How does the robustness of a genetic code influence adaptive landscape topography? To answer this question, we generated 100,000 rewired genetic codes by amino acid permutation, a rewiring scheme that preserves the synonymous codon block structure of the standard genetic code, but randomly permutes the 20 amino acids among these blocks [15,16] (Methods). We quantified the robustness of each code as the proportion of point mutations that do not change the physicochemical properties of amino acids, using the properties defined in ref. [22] (Fig C in S1 Text; Methods). According to this measure, the robustness of the standard genetic code is 0.385, meaning that 38.5% of point mutations do not change the physicochemical properties of amino acids. In comparison, the code robustness for the 100,000 rewired codes ranges from 0.257 to 0.462, with a median of 0.336, such that 5.48% of these codes exhibit robustness greater than or equal to the standard code. Therefore, when defining robustness in terms of multiple amino acid properties, the standard genetic code is highly robust, but not surprisingly so [15]. We then constructed an adaptive landscape using each of the 100,000 rewired genetic codes, for each of the 6 data sets, and characterized the topographies of these landscapes using three measures of landscape ruggedness [51]: the number of adaptive peaks, the prevalence of various types of epistasis, and the proportion of accessible mutational paths to the global peak (Methods). Whereas these 3 measures are all correlated with one another [52] (Table A in S1 Text), they illustrate different aspects of landscape topography, ranging from local (pairwise epistasis) to global (number and accessibility of adaptive peaks). Below, we focus our analyses on the GB1 data and report analogous results for the ParD, ParB, and DHFR data in Table B in S1 Text.

**Adaptive peaks.**   The number of adaptive peaks is a straightforward measure of landscape ruggedness, and thus of evolvability. The more local peaks a landscape has, the more likely an evolving population is to become trapped on one of these peaks, thus precluding the generation of further adaptive phenotypic variation. Under the standard genetic code, the GB1 landscape comprises 115 adaptive peaks, whereas under the 100,000 rewired codes, the number of adaptive peaks ranges from 97 to 478, with a median of 231.

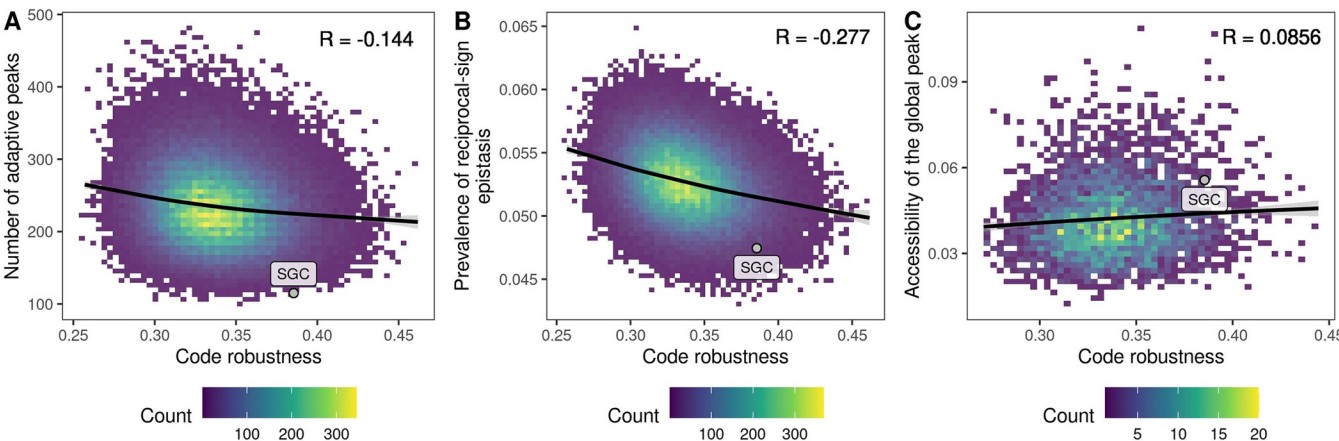

**Fig 2. More robust codes result in smoother adaptive landscapes.** Three measures of landscape ruggedness are shown in relation to code robustness, defined as the proportion of point mutations that do not change the physicochemical properties of amino acids. (A) The number of adaptive peaks, (B) the prevalence of reciprocal sign epistasis, and (C) the proportion of mutational paths to the global peak that are accessible. Panel (C) shows only genetic codes that preserve the size of the global peak relative to the standard genetic code and in which none of the amino acids contained in the global peak (WWLA) are encoded by the split codon block ($n = 3,769$). In each panel, the labeled point denotes the standard genetic code. All results pertain to the GB1 landscape. Analogous results for the ParD, ParB, and DHFR landscapes can be found in Table B in S1 Text. The data and code required to generate this Figure can be found at https://zenodo.org/records/10677993.

Fig 2A shows the number of adaptive peaks in relation to code robustness, revealing that more robust genetic codes tend to produce adaptive landscapes with fewer peaks than less robust codes (Pearson's correlation $R = -0.144$, $p < 2.2 \cdot 10^{-16}$). However, the trend is relatively weak, such that for any level of code robustness, there is considerable variation in the number of peaks. For example, for the 5.48% of codes with robustness greater than or equal to the standard code, the number of peaks ranges from 115 to 403. Strikingly, among all 100,000 codes, only 0.037% of the corresponding landscapes have less than or equal the number of peaks in the landscape produced by the standard code. The GB1 landscape is therefore exceptionally smooth under the standard genetic code. To a lesser extent, this is also true for the ParB-*parS* (2.3%) and ParB-*NBS* (4.6%) landscapes. However, this is not true for the ParD-ParE2, ParD-ParE3, and DHFR landscapes, where the number of peaks in the landscape under the standard genetic code lies in the 32.6%, 57.0%, and 20.5% quantile, respectively (Table C in S1 Text). Whether the standard genetic code is exceptional with regard to producing smooth fitness landscapes therefore appears to be data set-specific, a topic we return to later.

**Epistasis.** Epistasis, where a mutation's effect depends on the genetic background in which it occurs, is a cause of landscape ruggedness [53,54]. It can be classified into 3 types—magnitude, simple-sign, and reciprocal-sign [53] (Methods). Reciprocal-sign epistasis occurs when 2 mutations each have a positive (negative) effect on fitness, but each mutation has negative (positive) effect when introduced in the background of the other mutation. That is, the sign of each mutation's effect flips when introduced in the presence of the other mutation. Reciprocal-sign epistasis forms local valleys in an adaptive landscape, which preclude the generation of at least some adaptive phenotypic variation, thus decreasing evolvability. To measure the prevalence of these 3 types of pairwise epistasis, we randomly sample a large number of squares in each adaptive landscape's underlying mutational network, each of which contains an mRNA sequence variant, two of its single-mutant neighbors, and a double mutant that can be constructed from the single mutants. Based on the fitness values of these 4 sequences, we classify the type of epistasis the square exhibits (Methods).

Because more robust codes tend to produce adaptive landscapes with fewer adaptive peaks (Fig 2A), we expect landscapes produced under more robust codes to exhibit less reciprocal-sign epistasis than landscapes produced under less robust codes. Fig 2B confirms this expectation, showing a negative correlation between reciprocal-sign epistasis and code robustness ($R = -0.277$, $p < 2.2 \cdot 10^{-16}$). Similarly, simple-sign epistasis, which contributes to landscape ruggedness to a lesser extent than reciprocal-sign epistasis, because it involves only a single sign flip, also exhibits a negative correlation with code robustness (Table B in S1 Text). Robust genetic codes therefore diminish the kinds of epistatic interactions that cause landscape ruggedness.

**Global peak accessibility.** One consequence of landscape ruggedness is that the global adaptive peak may be less accessible to an evolving population, which may instead follow mutational paths to local adaptive peaks. We therefore expect that the global adaptive peaks of landscapes produced under more robust codes will be more accessible than those of landscapes produced under less robust codes. To test this, we quantified the mutational accessibility of the global peak of each landscape by calculating the probability that a randomly chosen, direct mutational path that starts at a randomly chosen mRNA sequence and ends at the global peak is accessible, meaning that fitness increases monotonically along the path [51,55,56]. In contrast to expectation, we observe that the global peaks of landscapes produced under more robust codes are not significantly more accessible for the ParD-ParE2 landscape ($R = 0.0052$, $p = 0.099$), and significantly less accessible for the ParD-ParE3 landscape ($R = -0.151$, $p < 2.2 \cdot 10^{-16}$).

We reasoned that the accessibility of the global peak might be confounded by its size: As the number of codons encoding an amino acid ranges from 1 to 6 in the amino acid permutation codes, the number of distinct mRNAs encoding a given protein variant can range from 1 to $6^L$, where $L$ is the number of sites in the protein. Indeed, we observe that the mutational accessibility of the global peak is strongly correlated with its size ($R = 0.801$, $p < 2.2 \cdot 10^{-16}$ for GB1; Fig D in S1 Text). Moreover, due to the fact that one of the synonymous codon blocks is split (UCN and AGY, with N denoting any nucleotide and Y denoting U or C; encoding serine in the standard genetic code), there might be several disconnected regions of the landscape encoding the protein sequence with the highest fitness value. When this is the case, the mutational accessibility of the global peak is significantly higher compared to codes where the global peak comprises a single connected region in genotype space (e.g., in the GB1 landscape, the mutational accessibility is 0.086 versus 0.055, $p < 2.2 \cdot 10^{-16}$, Welch two-sample $t$ test; Fig E in S1 Text). In order to make the landscapes more comparable, we restricted our analysis to only those landscapes in which the size of the global peak, in terms of number of mRNAs encoding the corresponding protein sequence, is the same as in the standard genetic code and, moreover, none of the amino acids contained in the global peak sequence are encoded by the split codon block. In this subset of landscapes, we observe the expected positive correlation between code robustness and accessibility of the global peak for all landscapes except for ParD-ParE3 (Fig 2C and Table B in S1 Text).

While statistically significant, the strength of the correlation between code robustness and global peak accessibility is weak and the magnitude of the effect is not large (mean global peak accessibility 0.055 versus 0.058 for the 1% least and most robust codes, respectively; Fig 2C). Given the low prevalence of sign epistatic interactions in the landscapes generated under even the least robust genetic codes, we reasoned that the range of landscape ruggedness observed in our data may simply be too small to observe a strong positive correlation between global peak accessibility and code robustness. This is indeed the case, as we confirmed by artificially inflating the ruggedness of the GB1 landscape (Section 1 in S1 Text).

In sum, the mutational accessibility of the global peak is strongly influenced by the number of its constituent mRNA sequences and whether they occupy disjoint regions of genotype space, and only weakly influenced by code robustness, due to the limited range of landscape ruggedness produced by the 100,000 amino acid permutation codes.

**Sensitivity analyses.** Our measure of code robustness assigns amino acids to discrete groups based on 7 key physicochemical properties, such as whether the amino acids are acidic or basic (Fig C in S1 Text; Methods). However, there are hundreds of physicochemical properties that can be used to characterize amino acids and the relevant amino acid properties can be protein- and site-specific. We thus repeated the analyses described above with code robustness defined in terms of each of the 553 properties from the AAindex database [57,58]. We show that amino acid properties generally either (1) do not influence landscape ruggedness significantly; or (2) influence it in the direction consistent with our previous results, i.e., increased robustness leads to decreased landscape ruggedness (Section 2 in S1 Text). We also see that the significant amino acid properties differ systematically among data sets (Section 2 in S1 Text).

To more clearly demonstrate the link between fitness-preserving mutations and landscape ruggedness, we also considered an alternative definition of code robustness, calculated as the expected fitness change upon mutation in a particular data set (Section 3 in S1 Text). Under this definition, robustness is not characterized by any particular physicochemical property of amino acids or by a combination of several such properties, but rather by which amino acids are interchangeable in a given data set. We again show that, when using this definition of robustness, robust codes lead on average to smoother fitness landscapes, with markedly stronger correlations than when using the aggregate measure of code robustness (Section 3 in S1 Text).

Finally, we show that our results are qualitatively insensitive to the way the rewired genetic codes are generated (Sections 4 and 5 in S1 Text) and to the dimensionality of the data set (i.e., the number of protein sites; Section 6 in S1 Text).

## Evolutionary simulations reveal complex relationship between code robustness and evolvability

Our analyses suggest that code robustness promotes evolvability by producing smooth adaptive landscapes with few peaks and little sign epistasis. As a consequence, we anticipate evolving populations to obtain higher fitness, on average, when translating proteins using more robust codes than when using less robust codes. To determine if this is the case, we turn to evolutionary simulations, specifically of greedy adaptive walks [7]. These model adaptive evolution of a large population with pervasive clonal interference, such that all possible point mutations to a sequence are simultaneously present in the population, and the fittest of these variants goes to fixation. For each of the 100,000 amino acid permutation landscapes and each of the 6 data sets, we initialized the walks in each nucleotide sequence encoding a functional product. We terminated a walk when it reached a local or global adaptive peak, and recorded the fitness of that peak sequence (Methods).

Fig 3A shows the average fitness reached by the greedy adaptive walks in relation to code robustness. As expected from our landscape-based analyses, evolving populations reached higher fitness, on average, when translating proteins using more robust genetic codes for the GB1, ParD-ParE2, and ParB-*NBS* landscapes (Table D in S1 Text). However, the results for the ParD-ParE3 landscape were not statistically significant ($R = -0.004$, $p = 0.182$) and we even observed a negative correlation in the ParB-*parS* ($R = -0.0118$, $p = 1.89 \cdot 10^{-4}$) and DHFR data sets ($R = -0.096$, $p < 2.2 \cdot 10^{-16}$). Similar to the analysis of accessible paths, we reasoned that these results might be caused by variation in the size of the global peak, such that larger global

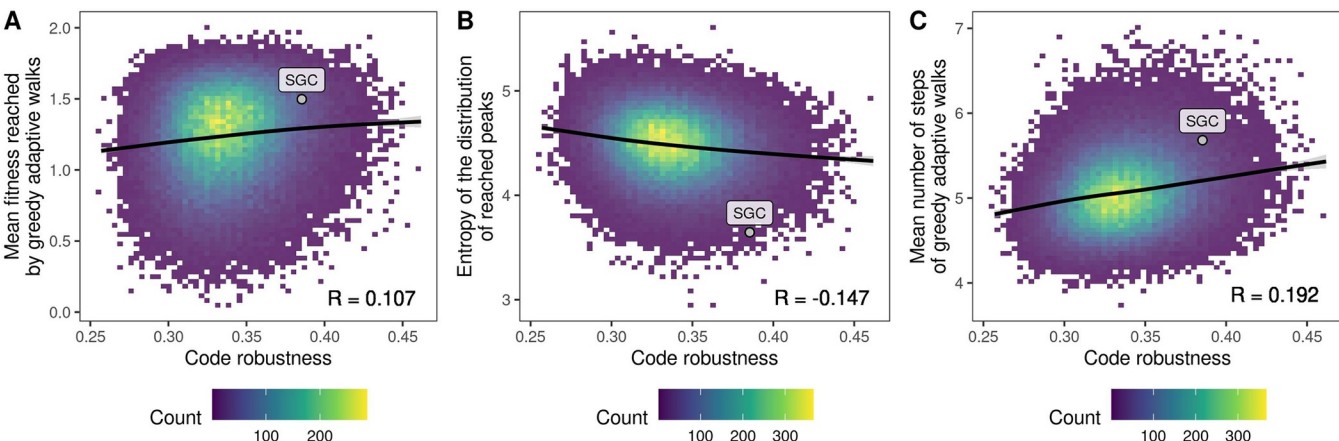

**Fig 3. Relationship between code robustness and results of greedy adaptive walks.** The labeled point denotes the results obtained using the standard genetic code. Data pertain to GB1. The data and code required to generate this figure can be found at https://zenodo.org/records/10677993.

peaks are easier to "find" than smaller global peaks, simply because they contain more mRNA sequences. Indeed, we observe a positive correlation between the size of the global peak and mean fitness reached by the greedy adaptive walks in all 6 data sets (Table E in S1 Text). We thus again restricted our analysis to those genetic codes for which the size of the global peak is the same as under the standard genetic code and occupies a single connected region in genotype space. However, even in this subset of codes, we observe a positive correlation between code robustness and mean fitness reached by the greedy adaptive walks in only for 4 out of 6 data sets (Table D in S1 Text). Whether code robustness promotes or diminishes evolvability thus depends on the particular landscape, which is surprising given that robust genetic codes are associated with smoother fitness landscapes in all 6 data sets. In Section 7 in S1 Text, we show that the correlations between code robustness and mean fitness result from a complex interplay between the heights and sizes of the basins of attraction of the peaks, as well as from idiosyncrasies specific to particular data sets.

We also observe that the average length of the walks tended to be longer under robust codes (Fig 3C; 4.89 versus 5.30 steps, on average, for the 1% least and most robust codes, respectively), revealing that the benefit of increased fitness afforded by code robustness comes at the cost of longer evolutionary trajectories to adaptation. This is in line with our observations concerning landscape ruggedness. In landscapes with many local peaks, a greedy walk is more likely to be initialized near one of these peaks, which it will likely ascend in only a small number of mutational steps. In contrast, in landscapes with few local peaks, a greedy walk is more likely to be initialized farther away from one of these peaks, thus increasing the length of the mutational path to adaptation, be it to a local or global peak.

We also highlight that the greedy walks reached higher mean fitness under the standard genetic code than under 85% of the amino acid permutation codes in 5 out of 6 landscapes and ranked among the top 5% in 3 of them. Similarly, the standard code resulted in exceptionally low Shannon entropy of the distribution of reached peaks in 5 out of 6 data sets (Table C in S1 Text), meaning that under the standard genetic code, greedy walks preferentially converged on a small number of fitness peaks. We observe qualitatively the same results in simulations of the weak-mutation regime (Section 8 in S1 Text) and using codes constructed by restricted amino acid permutation (Section 4 in S1 Text) and random codon assignment (Section 5 in S1 Text).

## The genetic code governs the genetic architecture of long-term molecular evolution

In the previous section, we studied a short-term adaptive process, in which high-fitness protein variants evolve from low-fitness variants via mutation and selection. However, once an evolving population reaches high fitness, it behaves like a random walk among the mutationally interconnected set of high-fitness variants. To assess how different code rewirings influence this random walk, we apply a visualization technique that captures the dynamics of a finite population evolving on a fitness landscape at mutation-selection-drift balance [59] where the distances between genotypes reflect the expected amount of time to evolve form one genotype to another (squared distances have units of time, and time is scaled such that each nucleotide mutation occurs at rate 1, see Methods).

In an earlier study, we used this technique to explore the structure of the GB1 landscape at the amino acid level [60] and found that it consists of 3 main regions of high-fitness protein variants that differ primarily in the placements of a small non-polar and bulkier amino acids at positions 41 and 54 (Fig F in S1 Text). The first and largest of these regions is characterized by having G at position 41, which is compatible with most amino acids at position 54 and contains the wild-type sequence (VDGV); we will refer to this as Region 1. The second region typically has G at position 54, while tolerating T at 54 in some contexts, together with L or F at position 41, and we will refer to this as Region 2. The final region, Region 3, is characterized by A at position 54, which can be paired at position 41 with C, S, or A, and to a lesser extent L and F. Moreover, each of these 3 regions is connected via functional intermediates with the other 2 regions (see Fig F in S1 Text and ref. [60]).

Here, we consider how the genetic code, standard or otherwise, reshapes the structure of these regions and restricts their mutational interconnections, focusing on the standard genetic code as well as the 2 most and 2 least robust in our set of 100,000 amino acid permutation codes (see Fig G in S1 Text for the corresponding codon tables). Fig 4 shows the resulting visualizations, where for each code we plot the visualization using a sufficient number of dimensions to show the major features of the corresponding fitness landscape. These dimensions are called Diffusion Axes because they reflect the dynamics of diffusion in sequence space, and they are ordered such that the first $k$ Diffusion Axes provide an optimal approximation of the expected times to evolve form one sequence to another (see Methods). In order to better see the structure of the high-fitness set, we also show a second visualization for each code where we only plot the high-fitness sequences (which in what follows we will take to be the fittest 1%) and color these sequences by their corresponding region in amino acid sequence space.

We find that different rewirings of the genetic code produce fitness landscapes with dramatically different structures from each other or from the structure of the fitness landscape in amino acid sequences space. For example, under the standard genetic code, Region 1 is no longer directly connected to Region 3 because neither 41F nor 41L is accessible from 41G under the standard genetic code (Fig 4A). Indeed, under many codes the set of high-fitness sequences becomes split into several distinct components separated by lower fitness sequences, as observed in Robust Code A (Fig 4B), Robust Code B (Fig H in S1 Text), and Non-Robust Code B (Fig 4E), so that moving from one component to another requires the fixation of less fit sequences. The waiting time for such deleterious fixations is long, increasing the amount of time required for a population to explore the landscape. We can quantify this in terms of the relaxation time of the rate matrix for the evolutionary random walk, given by the inverse of the absolute value of its largest non-zero eigenvalue. For Robust Code A and Non-robust Code B, the relaxation time is, respectively, 3.17 and 3.22-fold longer than the expected waiting time for individual nucleotide mutations; this is roughly 50% longer than for the standard genetic

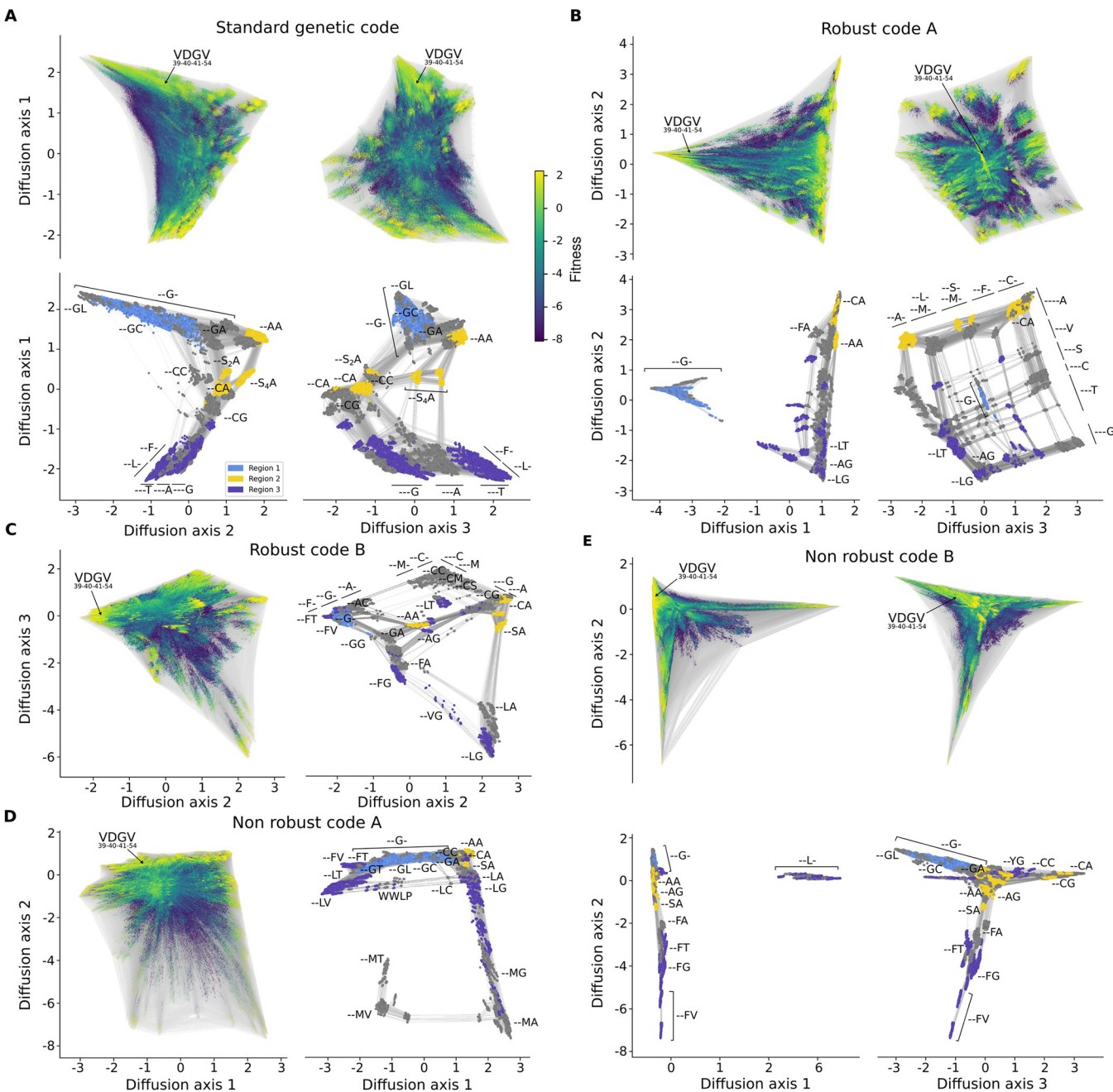

**Fig 4. The genetic code governs genotype network topology and the genetic architecture of long-term molecular evolution.** Fitness landscape for GB1 at positions 39, 40, 41, and 54 under the (A) standard genetic code, (B, C) the 2 most and (D, E) the 2 least robust codes in the amino acid permutation set. Vertices represent 12-nucleotide sequences and edges connect vertices if their corresponding sequences differ by a single point mutation. Vertex color represents protein fitness (color bar in (A) applies to all panels). Vertices are placed at the coordinates along the diffusion axes, which at a technical level are defined by the subdominant eigenvectors of the rate matrix describing the weak mutation dynamics and have units of square root of time [59], and where time is scaled such that each possible nucleotide mutation occurs at rate 1 (see Methods for details). For each pair of diffusion axes shown, there are 2 subpanels: one that shows all ≈16 million genotypes, with the location of the sequences encoding the wild-type protein sequence V39 D40 G41 V54 marked, and another that shows only the genotype network of high-fitness variants (top 1% of fitness distribution), which better shows the connectivity between high-fitness regions and which is annotated with the protein sequence features that characterize each cluster or subset of nucleotide sequences. Colors in these panels represent the main regions of functional protein sequences as highlighted in Fig F in S1 Text and show the extent to which the connectivity between these regions of amino acid sequence space is rewired under the different genetic codes. See Section 9 in S1 Text for a more detailed descriptions of the visualizations. The data and code required to generate this figure can be found at https://github.com/parizkh/rewired_codes_landscapes/tree/main/GB1/05_landscape_ visualizations.

code or Non-robust Code A with relaxation times of 2.31- and 2.05-fold longer than the expected waiting time for individual nucleotide mutations (see Methods for details).

We also find that high-fitness sequences connected by high-fitness paths can be connected via very different structures in sequence space. For example, the standard genetic code, as well as the 2 least robust codes (Fig 4A, D, and E) all show long branch-like structures where the high-fitness paths connecting genotypes require that the mutations be accumulated in a specific order. This can result in very long paths, for example, Fig I in S1 Text shows an example of an 11-mutation path connecting Regions 1 and 3 under the standard genetic code that does not include any substitutions at positions 39 or 40, synonymous changes, or reversions. In contrast, we can also see regions of sequence space where the high-fitness set shows a grid-like structure in which mutations at a pair of sites can accumulate independently from each other, as seen in the right-hand panel of Fig 4B for Robust code A, or under the standard genetic code, where a population can switch between F or L at position 41 more or less independently of whether T, A, or G is found at position 54 (Fig 4A, negative values of Diffusion Axis 1).

Besides these large-scale differences in the pattern of connectivity between high-fintess sequences, the density of high-fitness paths can also vary greatly. One particularly interesting case is where a pair of sequences are connected by many long high-fitness paths but are also accessible via a smaller number of short high-fitness paths that can only be accessed on very specific genetic backgrounds; we call these rare shortcuts "wormholes" because they are short paths that connect otherwise distant regions of the high-fitness set. For example, under the standard genetic code, we can see that distant parts of the network of high-fitness sequences are in fact accessible from one another via $41S_4$ (where the 2 disconnected sets of S codons are broken into $S_2$ and $S_4$, named for the number of codons in each set [61]; Fig 4A, bottom right). In this case, the average probability for 41G-54L sequences of arriving at a high-fitness genotype with L or F at 41 and T, A, or G at 54 through a high-fitness $S_4$ intermediate is 1.13% (see Section 9 in S1 Text for additional details). Thus, although these short paths are possible, they occur only a small minority of the time. We see another example of such a wormhole under Non-robust code A, where WWLP sequences bridge the otherwise distant regions characterized by 41L-54A and 41L-54T (Fig 4D). This wormhole is used even more rarely, only 0.002% of the time. In summary, these visualizations illustrate the richness and variety of landscape topographies that can be induced by different genetic codes, and the extent to which even exceptionally robust codes can interact with the idiosyncrasies of a particular protein fitness landscape to break crucial links between high-fitness variants.

## Codon compression schemes reveal additional code features influencing evolvability

Above, we have focused on amino acid permutation codes. However, engineering these codes in a living organism would require an extensive recoding of the genome, including the engineering of many orthogonal aminoacyl-tRNA synthetases and tRNAs. For example, in the most robust of the 100,000 codes we analyzed, only 2 amino acids occupy the same synonymous codon block as in the standard genetic code (Fig G in S1 Text, panel B). In contrast, to date, the synthetic biology community has engineered rewired genetic codes that change the meaning of up to only a small handful of codons [25,62–64]. There is therefore a large disconnect between the space of theoretically and practically realizable rewired genetic codes.

This motivated us to study a subset of rewired genetic codes that require only a small number of codon reassignments, as it may be possible to engineer these codes in a living organism using currently available technology. In particular, we studied the 57-codon *E. coli* genome synthesized by Ostrov and colleagues [23], in which all occurrences of 7 codons, from 4

synonymous codon blocks, together with the corresponding tRNAs were removed from the genome and are thus theoretically free for reassignment (Fig J in S1 Text). Assuming each of the 4 synonymous blocks is reassigned to one amino acid or a stop signal (as might be required by the tRNA wobble rules [65,66]), there are in total $21^4$ = 194,481 possible rewirings based on this compression scheme, one of them being the standard genetic code. We computationally generated all of these "Ostrov" codes and repeated the landscape-based analyses and evolutionary simulations described above.

Relative to the permutation codes, the Ostrov codes exhibited even stronger trends with respect to landscape ruggedness (Table F in S1 Text) and the outcomes of the greedy adaptive walks (Table G in S1 Text). Notably, the range of the landscape ruggedness measures, e.g., the number of peaks, is roughly the same as for the amino acid permutation codes, even though the Ostrov codes exhibit a much smaller range of code robustness (from 0.330 to 0.406, as compared to 0.257 to 0.462 for the permutation codes). Because the Ostrov codes differ from the permutation codes in that they do not all have the same synonymous codon block structure or the same number of stop codons as the standard code, we reasoned that these 2 structural features may provide an explanation for these observations.

The Ostrov codes can have more or fewer split codon blocks than the standard code. In the set of the 194,481 Ostrov codes, the number of split codon blocks ranges from 0 to 4 (Fig K in S1 Text). Increasing the number of split codon blocks decreases code robustness ($R$ = −0.345, $p$<2.2·10$^{-16}$; Fig L in S1 Text, Panel A), due to the increase in the number of non-synonymous mutations. This causes an increase in landscape ruggedness (Fig 5A and Table H in S1 Text), because maladaptive valleys can form in the mutational spaces between synonymous codons of split codon blocks. Consequently, the average fitness reached by the greedy adaptive walks consistently decreases as the number of split codon blocks increases in all data sets except for DHFR (Fig 5B and Table I in S1 Text). Consistent with the growing number of local peaks, we also observe that the adaptive walks get on average shorter and their endpoints less predictable as the number of split codon blocks increases (Table I in S1 Text). Interestingly, this effect remains even when restricting our analyses to codes that have the same robustness but differ in the number of split codon blocks (Figs M and N in S1 Text), showing that code robustness, as defined here, does not capture the full spectrum of effects mediated by changes in the number of split codon blocks.

The Ostrov codes can also have more or fewer stop codons that the standard code, which has 3 (UAG, UAA, and UGA). Because only the stop codon UAG has been freed for reassignment in the Ostrov codes, the minimum number of stop codons is 2, whereas the maximum is 9, corresponding to the assignment of all freed codons to a termination signal (Fig O in S1 Text). Panel B in Fig L in S1 Text shows that increasing the number of stop codons tends to decrease code robustness ($R$ = −0.160, $p$<2.2·10$^{-16}$), due to the increase in the number of nonsense mutations. Moreover, the number of stop codons is negatively correlated with the number of split codon blocks ($R$ = −0.269, $p$<2.2·10$^{-16}$), because if a codon block is assigned to a stop signal, it cannot be part of a split codon block. Thus, when measuring the effect of the number of stop codons on landscape ruggedness or the outcomes of adaptive walks, one has to condition on a given number of split codon blocks. In the following, we report results for codes with 0 split codon blocks; results for other numbers of split codon blocks can be found in Tables J and K in S1 Text. We observe that increasing the number of stop codons leads to an increase in the number of local peaks (Fig 5C and Table J in S1 Text), as well as decreased accessibility of the global peak (Table J in S1 Text); the effect on epistasis is more complex (Table J in S1 Text and Section 10 in S1 Text). Correspondingly, the average fitness reached by the greedy adaptive walks decreases as the number of stop codons increases (Fig 5D and Table K in S1 Text). This is expected, as in our adaptive landscapes, sequences containing stop codons are assigned a fitness value lower than any of the sequences without stop codons

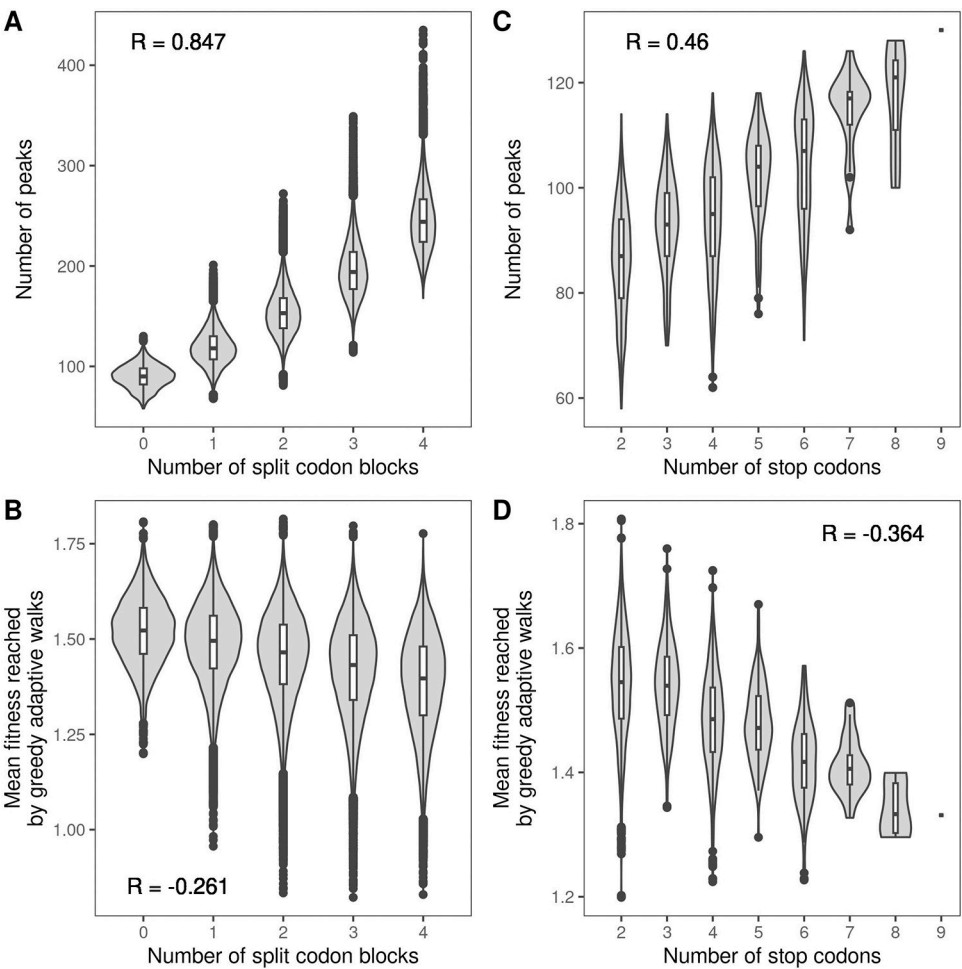

**Fig 5. Additional code features influencing protein evolvability.** Violin plots of the number of local peaks and the mean fitness reached by the greedy adaptive walks, shown in relation to (A, B) the number of split codon blocks in the 194,481 Ostrov codes and (C, D) the number of stop codons in the 3,965 Ostrov codes with no split codon blocks. Data pertain to GB1. The violin plots show the distribution and the box-and-whisker plots the median, 25th and 75th percentile. The upper whisker extends from the top of the box to the largest value no further than 1.5-times the inter-quartile range, the lower whisker extends from the bottom of the box to the smallest value no further than 1.5-time the inter-quartile range. Data beyond the end of the whiskers are plotted individually. The data and code required to generate this figure can be found at https://zenodo.org/records/10677993.

(Methods), reflecting the fact that the inclusion of a stop codon in an open reading frame causes the premature termination of translation and thus protein truncation, which is usually deleterious to protein function. We also observe that the greedy adaptive walks get shorter and less predictable as the number of stop codons increases (Table K in S1 Text). Moreover, these effects remain even among codes with the same robustness (Figs P and Q in S1 Text). In sum, our analyses of all possible code rewirings under the codon compression scheme proposed by Ostrov and colleagues [23] reveal additional code features influencing protein evolvability, namely the number of split codon blocks and the number of stop codons.

## Design principles: Genetic codes enhancing and diminishing evolvability

As discussed in Sections 2 and 3 in S1 Text, the amino acid properties most relevant for landscape topography are data set-dependent. This suggests that a genetic code that promotes

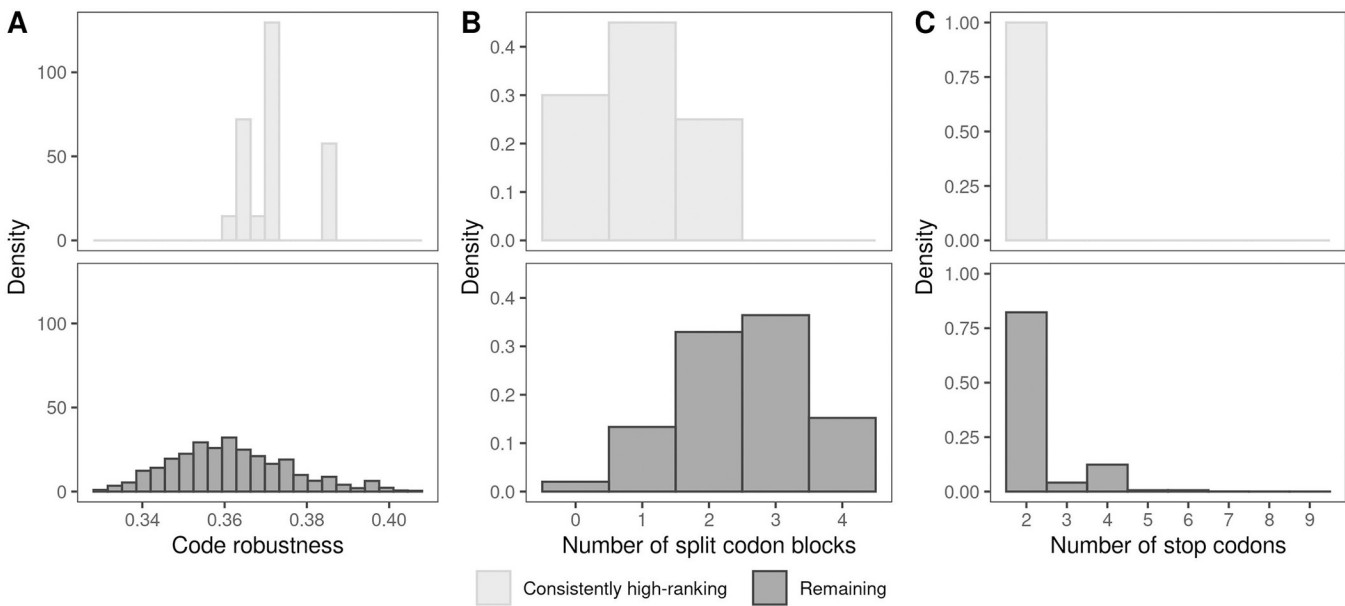

**Fig 6. Design principles for enhancing evolvability.** Comparison of the properties of the 20 consistently high-ranking (top 25%) codes (top row) with the remaining 194,461 codes (bottom row), in terms of (A) code robustness, (B) number of split codon blocks, and (C) number of stop codons. The data and code required to generate this figure can be found at https://zenodo.org/records/10677993.

evolvability for one protein might not do so for another. Indeed, in our evolutionary simulations with the Ostrov codes, the mean fitness reached by the greedy walks is not strongly correlated across our 6 data sets (Table L in S1 Text). Nonetheless, there is a small subset of codes that promote evolvability across all 6 data sets, and we reasoned that these may exhibit commonalities that could inform design principles for engineering genetic codes to promote evolvability across a diversity of proteins.

We therefore ranked each Ostrov code in descending order according to mean fitness reached in the evolutionary simulations, separately for each of the 6 data sets. There are 20 codes that consistently rank in the top 25% of all 6 lists. We note that the standard genetic code is not a member of this set of consistently high-ranking codes, as it does not rank in the top 25% of codes for the GB1, ParD-ParE3, and DHFR data sets (Table M in S1 Text). This shows that even relatively small changes to the standard genetic code can enhance protein evolvability.

We then compared these consistently high-ranking codes to the remaining 194,461 codes, in terms of robustness, number of split codon blocks, and number of stop codons. The consistently high-ranking codes have significantly higher robustness ($p = 1.06 \cdot 10^{-5}$, Welch two-sample $t$ test; Fig 6A), fewer split codon blocks ($p = 2.35 \cdot 10^{-8}$, Welch two-sample $t$ test; Fig 6B), and fewer stop codons ($p < 2.2 \cdot 10^{-16}$, Welch two-sample $t$ test; Fig 6C). Similarly, for the 280 genetic codes that consistently rank among the bottom 25% of codes, we observe significantly lower robustness ($p < 2.2 \cdot 10^{-16}$, Welch two-sample $t$ test) and more split codon blocks ($p < 2.2 \cdot 10^{-16}$, Welch two-sample $t$ test) compared to the remaining codes; however, the consistently low-ranking codes do not tend to have more stop codons ($p = 1$, Welch two-sample $t$ test) (Fig R in S1 Text). This suggests that there are some basic design principles to engineering genetic codes that promote evolvability across a diversity of proteins. Specifically, minimize the number of split codon blocks, minimize the number of stop codons, and assign amino acids to codon blocks such that point mutations cause only small changes to amino acid

properties, using an aggregate measure of a diversity of amino acid properties [22]. The opposite is true for codes that diminish evolvability, perhaps with the exception of the number of stop codons. To illustrate these design principles, in Section 11 in S1 Text, we give examples of Ostrov codes that, in our simulations, consistently decrease or increase evolvability, and compare them with codes specifically designed to promote [22] or diminish [34] evolvability.

## Discussion

The standard genetic code defines the rules of protein synthesis for nearly every life form on Earth [67]. It imparts an extreme, "one in a million" level of error tolerance [16] that buffers the deleterious effects of infidelity in replication, transcription, and translation [15,16], and provides a striking example of biological robustness at the heart of an essential cellular information processing system [68]. However, prior theoretical work, limited by a lack of suitable data, has disagreed on whether such robustness hinders [22] or facilitates [19] protein evolvability. Here, by computationally translating millions of mRNA sequences under hundreds of thousands of rewired genetic codes using experimental data for 6 proteins, we reveal that code robustness facilitates protein evolvability by rendering smooth adaptive landscapes upon which evolving populations readily find mutational paths to adaptation. At the same time, our results suggest that the standard genetic code is likely not "one in a million" with respect to evolvability. Moreover, whereas the correlations we observe between code robustness and landscape ruggedness are consistent in direction across data sets, they are relatively weak, so that genetic codes with a similar degree of robustness may differ substantially in the degree of evolvability they confer.

Landscape ruggedness has long been viewed as an impediment to adaptation [1] and, as such, has been used as a proxy for evolvability [4]. The intuition is that ruggedness frustrates evolvability by blocking "uphill" mutational paths to the global adaptive peak, thus limiting the ability of mutation to bring forth adaptive phenotypic variation. Our results confirm this intuition in the context of rewired genetic codes, in that adaptive walks tend to achieve higher fitness on smooth landscapes caused by robust codes than on rugged landscapes caused by less robust codes. However, this is not solely attributable to an increase in the mutational accessibility of the global peak. Rather, as ruggedness decreases, a positive correlation emerges between the height of a peak and the size of its basin of attraction. This causes adaptive walks to preferentially converge on a small number of high-fitness peaks in less rugged landscapes, and more uniformly to all peaks in more rugged landscapes. Moreover, as illustrated by the DHFR data set (Section 7.1 in S1 Text), smooth landscapes caused by robust codes may comprise fitness peaks that are lower than the fitness peaks of rugged landscapes caused by non-robust codes, such that increased landscape ruggedness is actually associated with increased protein evolvability. As such, simplistic measures of landscape ruggedness based solely on the number of peaks may be an insufficient proxy for evolvability [4] or for predicting evolutionary dynamics [69].

An additional factor influencing the accessibility of the global adaptive peak is its size. By randomly permuting amino acids among synonymous codon blocks, we created landscapes that vary significantly in the number of mRNA sequences that translate to the highest-fitness protein sequence. In comparing the outcomes of evolutionary simulations on these landscapes, we observe that protein sequences encoded by a large number of mRNA sequences are easier to evolve than equally fit sequences encoded by fewer mRNA sequences [70–72]. This observation implies that amino acids encoded by a large number of codons, such as serine or leucine, should be relatively more abundant in protein sequences than amino acids encoded by few codons, such as methionine or tryptophan. Indeed, across the tree of life, there is a positive

correlation between the abundance of an amino acid and its number of constituent codons [73,74], and as early as 1973, Jack L. King attributed this correlation to differences in amino acid "findabilities" caused by the structure of the standard genetic code [75]. Our results generalize this observation to nonstandard genetic codes, and suggest that if life had converged on a different standard code, the amino acid composition of proteins would likely be very different from the one we know. Such variation in the proteomic abundance of amino acids may already be apparent in the proteomes of organelles and organisms that use nonstandard genetic codes in nature [67,76,77], and may emerge in directed laboratory evolution experiments that use synthetic organisms with nonstandard genetic codes [24,25]. If so, these systems may provide empirical support for entropic arguments regarding adaptation [71,72].

There are several caveats to the results presented here. First, because there are so few combinatorially complete data sets measuring a quantitative phenotype for all $20^L$ protein variants, our conclusions are based on only 6 empirical adaptive landscapes. Moreover, the 6 landscapes differ in the strength of the relationship between code robustness and landscape ruggedness (see, e.g., Table B in S1 Text) and, in some cases, even in the sign of the trend. We believe this results from the low dimensionality of our landscapes, in that the landscapes are largely influenced by idiosyncrasies specific to a single protein site [78,79]. For example, we observe that the fitness reached by greedy walks in the DHFR landscape is mostly determined by whether the genetic code allows a mutation from cysteine to aspartate or glutamate (Section 7.1 in S1 Text). Indeed, our ablation analysis of the GB1 landscape suggests that the average strength of the correlation between robustness and evolvability increases as the number of protein sites increases from 2 to 4 (Section 6 in S1 Text). Moreover, the data sets we use differ in several important aspects—the level of ruggedness of the corresponding landscape under the standard code, the location of the screened residues in the protein, as well as the assayed phenotype—all of which support the generality of our findings. Second, for three of our data sets, the measured phenotype is the relative binding affinity of the protein to its ligand, and it is not immediately apparent how this phenotype relates to organismal fitness. Even so, a large body of literature attests to the power of such quantitative phenotypes in teaching us about protein evolvability [48]. Third, in the main text, we defined code robustness using a discrete categorization of amino acids based on their physicochemical properties (Fig C in S1 Text). A mutation between 2 amino acids belonging to the same physicochemical group was considered "synonymous." While it is well known that some amino acid pairs are more exchangeable than others [80–82], it is a simplification to assume that these amino acids are entirely equivalent. To overcome this limitation, we additionally considered definitions of code robustness based on a large number of diverse physicochemical properties, one at a time (Section 2 in S1 Text), and on a custom amino acid similarity score specific to each of the 6 data sets (Section 3 in S1 Text), with qualitatively the same results. Fourth, the error tolerance of a genetic code, standard or nonstandard, is influenced by mutation bias and codon usage [16,83] as they make some mutations more likely than others. While mutation bias and codon usage may influence peak accessibility in adaptive landscapes [84], they do not affect landscape topography, which is why we have not considered these effects here. We hope that in the future, it will become possible to overcome these caveats and confirm our results, both theoretically, as more and larger combinatorially complete data sets become available, and experimentally, by comparing the dynamics and outcomes of laboratory evolution experiments with proteins and organisms that use different genetic codes.

Such experiments are becoming more broadly accessible, as a diversity of recoded organisms and plasmid-borne orthogonal translation systems are now commercially available. Moreover, these experiments are becoming increasingly scalable. For example, Zürcher and colleagues [25] have recently engineered bacterial strains with as many as 16 different genetic

codes. Understanding the relationship between code structure and evolvability is therefore highly topical, as the future in which synthetic organisms with nonstandard genetic codes are utilized in science and in industry [85,86], for example, to accelerate directed evolution experiments [3,22,25], to achieve bio-containment [25,34,64,87–89], or to produce drugs [90–92] is now tangibly close. We have identified general design principles, as well as a few concrete candidate codes, that are expected to increase evolvability beyond that of the standard genetic code. We have identified even more genetic codes that decrease evolvability, which may be useful for the bio-containment of synthetic organisms. All of these codes are compatible with the 57-codon *E. coli* genome reported by Ostrov and colleagues [23], and could thus be engineered in the lab using existing technology. Our analyses with this codon compression scheme explored all 194,481 genetic codes that reassign one or more of the freed codon blocks, assuming that the whole synonymous codon block needs to be assigned to one amino acid. If this assumption is lifted [25,93], it would, in the context of the 57-codon *E. coli* genome, lead to a staggering $21^7 \approx 1.8 \cdot 10^9$ possible code rewirings. Together with other codon compression schemes [26,94–96], the space of possible code rewirings available today is practically infinite and will continue to grow as larger-scale rewirings become feasible.

Building upon our results, there are several directions for future research. First, the advances in biotechnology discussed above now enable experimental tests of the relationship between genetic code robustness and evolvability, as recently proposed by Zürcher and colleagues [25]. Do robust genetic codes indeed lead to larger improvements of protein function in directed evolution experiments? And are organisms with robust genetic codes better able to adapt to changing environmental conditions? Second, in this paper we have worked only with rewired genetic codes, i.e., codes that change the mapping between codons and amino acids, but we have not considered expanded genetic codes, i.e., codes that include a 21st, nonstandard amino acid. While a small number of evolutionary experiments using organisms with expanded genetic codes have been reported [97–100], how the addition of a 21st nonstandard amino acid influences protein and organismal evolvability is not yet fully understood. This question could be addressed experimentally by generating combinatorially complete data for all $21^L$ sequence variants, using a diversity of 21st nonstandard amino acids, and also theoretically, for example by subsampling combinatorially complete data to contain fewer than 20 amino acids [101].

## Materials and methods

### Data processing

We estimated the fitness of each measured amino acid variant following Rubin and colleagues [102].

For each replicate experiment $i$, the fitness of variant $v$ is equal to

$$f_{v,i} = \log\left(\frac{c_{v,i,\text{sel}} + \frac{1}{2}}{c_{wt,i,\text{sel}} + \frac{1}{2}}\right) - \log\left(\frac{c_{v,i,\text{inp}} + \frac{1}{2}}{c_{wt,i,\text{inp}} + \frac{1}{2}}\right),$$

where $c_{v,i,\text{sel}}$ is the count of variant $v$ in the $i$-th replicate sample after selection, $c_{wt,i,\text{sel}}$ is the count of the wild type in the $i$-th replicate sample after selection, $c_{v,i,\text{inp}}$ is the count of variant $v$ in the $i$-th replicate input sample, and $c_{wt,i,\text{inp}}$ is the count of the wild type in the $i$-th replicate

input sample. The variance of the estimate is equal to

$$\sigma_{v,i}^2 = \frac{1}{c_{v,i,\text{inp}} + \frac{1}{2}} + \frac{1}{c_{v,i,\text{sel}} + \frac{1}{2}} + \frac{1}{c_{wt,i,\text{inp}} + \frac{1}{2}} + \frac{1}{c_{wt,i,\text{sel}} + \frac{1}{2}}.$$

The final fitness of variant $v$ is then the weighted average of the $r$ replicates, with weights given by the inverse of the corresponding variance:

$$f_v = \frac{\sum_{i=1}^{r} \frac{1}{\sigma_{v,i}^2} f_{v,i}}{\sum_{i=1}^{r} \frac{1}{\sigma_{v,i}^2}}$$

and the variance is computed as follows:

$$\sigma_v^2 = \frac{1}{\sum_{i=1}^{r} \frac{1}{\sigma_{v,i}^2}}.$$

The number of replicates $r$ equals 1 for GB1, ParB-*parS*, and ParB-*NBS*; 2 for ParD-ParE2 and ParD-ParE3; and 6 for DHFR.

In the ParB-*parS* and ParB-*NBS* data sets, read counts of 79,187 variants were not reported in the original publication. It is those variants that had 0 reads in both post-selection samples. When computing the fitness and variance for these variants, we assumed the input read counts are equal to the median input read count over all protein variants with 0 reads after selection for binding a given DNA-motif.

The DHFR landscape was measured on the level of codons, i.e., it contains a fitness measurement for each synonymous encoding of a given protein variant. Because any connection between the original codons and the amino acids is lost in our randomized genetic codes, we assume, as for the other landscapes, that all synonymous variants have the same fitness. This protein-level fitness is computed by first pooling the reads based on the encoded amino acids and then proceeding as described above. In the original study, the bottom 93% of the nucleotide variants, according to fitness, were considered nonfunctional [40]; we assume the same for the 7,173 corresponding protein variants (89.7% of the landscape). We assigned these variants the same fitness as variants containing stop codons (see below).

Based on these raw fitness estimates and variances for observed variants, we imputed the fitness values for the missing variants and inferred the full adaptive landscape as the maximum a posteriori estimate under empirical variance component regression, an empirical Bayes modeling framework that naturally incorporates all orders of genetic interaction [49].

## Constructing adaptive landscapes

To construct an adaptive landscape, we consider the set of all possible mRNA sequences of length 12 (for GB1, ParB-*parS*, and ParB-*NBS*, so that they encode 4 amino acids; there is $4^{12} = 16,777,216$ such sequences) or 9 (for ParD-ParE2, ParD-ParE3, and DHFR, encoding 3 amino acids; $4^9 = 262,144$ sequences), respectively. We represent each sequence as a vertex in a mutational network. Two vertices are connected with an edge if the Hamming distance of the corresponding mRNA sequences is 1, i.e., the 2 sequences differ by a single point mutation. This underlying network is the same for all genetic codes.

For each genetic code, we then assign an "elevation" to each vertex, equal to the fitness of the sequence, translated using a given genetic code. Sequences containing stop codons are assigned an arbitrary elevation lower than the fitness of any sequence not containing stop codons (we used a value of −100, but the precise value is not relevant for the analyses presented here).

## Generating rewired genetic codes

The amino acid permutation codes were generated by randomly permuting the 20 amino acids among the synonymous codon blocks. The number and position of stop codons were fixed as in the standard genetic code, as well as the presence of exactly 1 split codon block (i.e., the UCN and AGY codons, encoding serine in the standard genetic code, always encode the same amino acid). There is $20! \approx 2.4 \cdot 10^{18}$ such codes, of which we randomly sampled 100,000.

For the Ostrov codes, we assumed that each of the freed synonymous codon blocks (UUA +UUG; UAG; AGU+AGC; AGA+AGG) is assigned one of the 20 amino acids or a stop signal. All other codons retain their meaning as in the standard genetic code. In total, there is $21^4 = 194{,}481$ such codes, one of them being the standard genetic code. We generated all of these genetic codes.

## Code robustness

We define code robustness as the proportion of single-nucleotide substitutions that do not change the physicochemical properties of amino acids. We divided amino acids into 7 physico-chemical groups following Pines and colleagues [22] (Fig C in S1 Text): acidic (D, E); aliphatic (A, I, L, V); aromatic (F, W, Y); basic (H, K, R); glycine (G); polar (C, M, N, Q, S, T); and pro-line (P). We considered mutations from an amino acid to a stop codon or vice versa as a change in physicochemical properties, whereas we did not consider mutations among stop codons as a change in physicochemical properties.

## Number of adaptive peaks

Intuitively, a local peak is a sequence whose fitness is higher than the fitness of any of its neighbors. In our case, due to the degeneracy of the genetic code, several vertices often have the same elevation and, moreover, those vertices will usually be connected; peaks are thus usually plateaus rather than a single vertex. Formally, we define a local peak as a set of vertices that (1) are connected in the genotype space; (2) all have the same elevation; and (3) whose neighbors are either part of the set or have a lower elevation.

## Epistasis analysis

A square is a quadruplet of sequences that contains a "wild type" sequence, 2 of its one-mutant neighbors, and the corresponding double mutant. In the following, we denote by $f_{00}$ the fitness of the wild type, by $f_{01}$ and $f_{10}$ the fitness of the 2 single mutants, and by $f_{11}$ the fitness of the double mutant. The "mutational effect" of a given mutation is denoted by $\Delta f$, e.g., $\Delta f_{00 \to 10} = f_{10} - f_{00}$ is the change in fitness caused by mutating the wild-type sequence to one of the single mutants. We say that there is no epistasis if

$$f_{00} + f_{11} - f_{01} - f_{10} = 0.$$

The square is classified as having magnitude epistasis if

$$\Delta f_{00 \to 10} \cdot \Delta f_{01 \to 11} > 0 \text{ and } \Delta f_{00 \to 01} \cdot \Delta f_{10 \to 11} > 0,$$

i.e., the effects of both mutations have the same sign (increase fitness or decrease fitness) regardless of the genetic background. Similarly, the square is classified to have reciprocal-sign epistasis if

$$\Delta f_{00 \to 10} \cdot \Delta f_{01 \to 11} < 0 \text{ and } \Delta f_{00 \to 01} \cdot \Delta f_{10 \to 11} < 0,$$

i.e., the effects of both mutations have opposite signs in different genetic backgrounds. The

remaining cases, i.e.,

$$\Delta f_{00\to10} \cdot \Delta f_{01\to11} > 0 \text{ and } \Delta f_{00\to01} \cdot \Delta f_{10\to11} < 0$$

and

$$\Delta f_{00\to10} \cdot \Delta f_{01\to11} < 0 \text{ and } \Delta f_{00\to01} \cdot \Delta f_{10\to11} > 0$$

are classified as simple-sign epistasis: the sign of one of the mutations is the same in the different backgrounds, whereas the sign of the second mutation changes in the different backgrounds.

Due to the size of the genotype networks, listing all squares is computationally prohibitive. Thus, we randomly sampled 1,000,000 squares by first sampling a random sequence and then sampling 2 random mutations at 2 different positions in the sequence.

## Mutational accessibility of the global peak

We define the mutational accessibility of the global peak as the probability that, picking a random functional sequence and a random direct path from the sequence to the global peak, the chosen path is accessible, i.e., the fitness increases monotonically along the path. In our case, the global peak is composed of several mRNA sequences; we define direct paths as those paths that reach any of the global peak sequences in the smallest possible number of steps. For example, considering the standard genetic code and the ParD-ParE3 data set, the global peak consists of 4 sequences: GAU UGG GAA, GAU UGG GAG, GAC UGG GAA, and GAC UGG GAG (all translate to DWE). Starting from sequence GAU UGG AUG (DWM), there are 2 direct paths to the global peak: GAU UGG AUG—GAU UGG **G**UG—GAU UGG G**A**G and GAU UGG AUG—GAU UGG A**A**G—GAU UGG **G**AG. Notice that in both cases, the end point of the direct paths is only one of the 4 global peak sequences, since reaching the other 3 sequences in the global peak would require more than 2 mutations (and hence those paths are not considered direct).

## Greedy adaptive walks

We simulated greedy adaptive walks on the landscape in which the most fit of the 1-mutant neighbors is fixed in every step, until a global or local peak is reached. However, the degeneracy of the genetic code means that the fitness values in the landscape are not unique, as all mRNA sequences encoding the same protein share the same fitness. The "most fit" neighbor thus does not have to be uniquely defined, e.g., because there are several possible mutations that lead to the same fitness increase, or because a neutral plateau must be crossed before new adaptive variants may be generated. If this happens, we retain all sequences with the highest fitness; we then explore all of their 1-mutant neighbors and choose the fittest one(s) of those, etc.

We initiated the walks in all possible functional sequences.

## Entropy of the distribution of reached peaks

For the greedy adaptive walks, we compute the entropy of the walks' targets as follows:

$$-\sum_{v\in\mathcal{V}} P(v)\log P(v),$$

where $\mathcal{V}$ is the set of all endpoints of the greedy walks (i.e., the set of all adaptive peaks) and $P(v)$ denotes the proportion of greedy walks that terminate on peak $v$.

## Visualization of the GB1 landscape under rewired genetic codes

We used the visualization method as previously described [59]. Briefly, we construct a model of molecular evolution where a population evolves via single nucleotide substitutions and the rate at which each possible substitution becomes fixed in the population is related to its relative selective advantage or disadvantage. Specifically, the rate of evolution from sequence $i$ to any mutationally adjacent sequence $j$ is given by

$$Q_{ij} = \frac{S_{ij}}{1 - e^{-S_{ij}}}$$

where $S_{ij}$ is the scaled selection coefficient (population size times the selection coefficient of $j$ relative to $i$) and the total leaving rate from each sequence $i$ is given by

$$Q_{ii} = -\sum_j Q_{ij}.$$

In this context, we assume that the selection coefficient between sequences $i$ and $j$ is proportional to the difference in log-enrichment scores or fitness $f_j - f_i$ and therefore $S_{ij} = c(f_j - f_i)$, where $c$ controls the strength of selection. For all analyses presented here, we used the simple choice of c = 1, which for the standard genetic code gives a mean fitness at stationarity equal to 0.055, similar to the wild-type sequence V39 D40 G41 V54 (which in this experiment by definition has fitness 0). Given the rate matrix $Q$, we then construct the visualization by using the subdominant right eigenvectors $r_k$ associated with the smallest magnitude non-zero eigenvalues $\lambda_k$ of this rate matrix as coordinates for the low dimensional representation of the landscape, where each such coordinate defines one of the "diffusion axes" used in the visualization. Because the smallest magnitude non-zero eigenvalues and their associated eigenvectors control the most slowly decaying deviations from the stationary distribution, the resulting visualization reflects the long-term barriers to diffusion in sequence space and clusters in the representation correspond to sets of initial states from which the evolutionary model approaches its stationary distribution in the same way. Thus, multi-peaked fitness landscapes appear as broadly separated clusters with 1 peak in each cluster. Moreover, by scaling the axes appropriately, as is done here,

$$u_k = \frac{r_k}{\sqrt{-\lambda_k}},$$

these axes $u_k$ have units of square-root of time, where time is measured in the expected number of neutral substitutions for a completely neutral sequence. In particular, using these coordinates $u_k$, the squared Euclidean distance between arbitrary sequences $i$ and $j$ equals the sum of the expected time $H_{ij}$ to evolve from $i$ to $j$ and the expected time $H_{ji}$ to evolve from $j$ to $i$, i.e.:

$$\sum_k \left( u_{k,i} - u_{k,j} \right)^2 = H_{ij} + H_{ji}.$$

Using the first several $u_k$ (i.e., $u_1$ and $u_2$ for a 2D representation or $u_1$, $u_2$, and $u_3$ for a 3D representation) optimally preserves the above relation in a principal components sense (see ref. [59] for details). Moreover, the associated eigenvalues $\lambda_k$ reflect the negative rate at which populations diffuse across these barriers, so that $-\frac{1}{\lambda_k}$ gives the timescale it would take to overcome the barriers captured by $u_k$. Thus, the longest of these timescales, $-\frac{1}{\lambda_1}$ can be used to characterize how fast a population explores a given fitness landscape. Indeed, $-\frac{1}{\lambda_1}$ is also known as the relaxation time of the Markov chain and is closely linked to many other metrics for describing how quickly a Markov chain approaches its stationary distribution [103].

Additionally, we calculated the probability of using the wormholes under this evolutionary model using standard Markov chain theory [104]. Specifically, we can compute the probability of arriving at a set of genotypes B from another A through specific mutations by defining an absorbing Markov chain in which each mutation entering B is replaced by a corresponding absorbing state, and where the transition rate into this absorbing state is the same rate as the corresponding transition rate in the original chain. We then split these absorbing states into 2 subclasses B1, corresponding to our transitions of interest, and B2 corresponding to the other transitions into B. We calculate the probability of being absorbed into B1 for each genotype in A and take the average over all genotypes in A.

## Supporting information

**S1 Text. Supplementary figures, tables, and texts.**
(PDF)

## Acknowledgments

We thank Andreas Wagner and Macarena Toll-Riera for discussions, Václav Rozhoň for help with algorithm design and implementation, and Thuy-Lan Lite for kindly providing the counts data needed for the construction of the ParD-ParE2 and ParD-ParE3 landscapes.

## Author Contributions

**Conceptualization:** Hana Rozhoňová, Joshua L. Payne.

**Data curation:** Hana Rozhoňová.

**Formal analysis:** Hana Rozhoňová, Carlos Martí-Gómez, David M. McCandlish, Joshua L. Payne.

**Funding acquisition:** Joshua L. Payne.

**Investigation:** Hana Rozhoňová, Carlos Martí-Gómez, David M. McCandlish, Joshua L. Payne.

**Methodology:** Hana Rozhoňová, Carlos Martí-Gómez, David M. McCandlish, Joshua L. Payne.

**Project administration:** David M. McCandlish, Joshua L. Payne.

**Resources:** David M. McCandlish, Joshua L. Payne.

**Software:** Hana Rozhoňová, Carlos Martí-Gómez, David M. McCandlish.

**Supervision:** David M. McCandlish, Joshua L. Payne.

**Validation:** Hana Rozhoňová.

**Visualization:** Hana Rozhoňová, Carlos Martí-Gómez, David M. McCandlish, Joshua L. Payne.

**Writing – original draft:** Hana Rozhoňová, Joshua L. Payne.

**Writing – review & editing:** Hana Rozhoňová, Carlos Martí-Gómez, David M. McCandlish, Joshua L. Payne.

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
