## [Editor Report · Decision Letter 0]

17 Jul 2023

Dear Dr Payne, 

Thank you for submitting your manuscript entitled "Protein evolvability under rewired genetic codes" for consideration as a Research Article by PLOS Biology.

Your manuscript has now been evaluated by the PLOS Biology editorial staff (this was greatly helped by the fact that I saw your excellent talk at the EMBO meeting last week!), as well as by an academic editor with relevant expertise, and I'm writing to let you know that we would like to send your submission out for external peer review.

Once your full submission is complete, your paper will undergo a series of checks in preparation for peer review. After your manuscript has passed the checks it will be sent out for review. To provide the metadata for your submission, please Login to Editorial Manager (https://www.editorialmanager.com/pbiology) within two working days, i.e. by Jul 19 2023 11:59PM.

Kind regards,

Roli Roberts

Roland Roberts, PhD

Senior Editor

PLOS Biology

rroberts@plos.org

---

## [Decision Letter · Decision Letter 1]

31 Aug 2023

Dear Dr Payne,

Thank you for your patience while your manuscript "Protein evolvability under rewired genetic codes" was peer-reviewed at PLOS Biology. It has now been evaluated by the PLOS Biology editors, an Academic Editor with relevant expertise, and by four independent reviewers. 

You'll see that the reviewers are broadly positive (though reviewer #3 is somewhat sceptical regarding the magnitude of advance), but three of them raise a number of concerns that must be addressed before we can consider it further. In light of the reviews, which you will find at the end of this email, we would like to invite you to revise the work to thoroughly address the reviewers' reports.

IMPORTANT: We discussed the reviewers' comments with the Academic Editor, who sent the following additional guidance that you might find helpful when deciding how to proceed with your revisions:

(1) Shorten the manuscript. I agree with reviewer 1 that it is far too long, and kind of meanders around, especially in the second half of the results. But I also think that it is very well written and clearly explained, and it would be a disservice to ask the authors to reduce the explanatory text. Hence, I like the idea of removing a part of the results altogether, or perhaps moving some things to the SI and only briefly mentioning it in the main text. E.g. the network analysis adds relatively little value but takes up a lot of space, as do sections 2.6 and 2.7.

(2) Address the impact of the method used to calculate robustness and clarify some of the methods that are missing or unclear (as suggested by reviewer 2). Which physicochemical properties were used, and which ones were most important? Also, on line 158 there's a mention of imputation of fitness landscapes and missing sequence variants, but there are no details about this in the methods. I am not sure exactly what was missing, how it was imputed, and whether this impacts the fitness landscape and/or the outcomes.

(3) Clearly discuss the very weak correlations and different outcomes for the different proteins, and resulting biological implications, as indicated by reviewer 3. I think these issues do reduce the novelty and impact of the results; at the very least they should be clearly discussed. The low dimensionality of the dataset (reviewer 4) is similarly an important concern that should be discussed.

Given the extent of revision needed, we cannot make a decision about publication until we have seen the revised manuscript and your response to the reviewers' comments. Your revised manuscript is likely to be sent for further evaluation by all or a subset of the reviewers.

**IMPORTANT - SUBMITTING YOUR REVISION**

*Re-submission Checklist*

*Published Peer Review*

*PLOS Data Policy*

*Blot and Gel Data Policy*

Sincerely,

Roli Roberts

Roland Roberts, PhD

Senior Editor

PLOS Biology

rroberts@plos.org

REVIEWERS' COMMENTS:

Reviewer #1:

[identifies himself as Stephen Freeland]

Protein evolvability under rewired genetic codes

Hana Rozhoňová, Carlos Martí-Gómez, David M. McCandlish and Joshua L. Payne

Before delving into the intellectual content of this manuscript, let me state how well written it is - well organized and with a clarity of written English that surpasses most manuscripts I review. As a reviewer, this quality of preparation makes a real difference to how easily I can assess the content. As a potential reader, this contributes directly to how likely I am to actually read the work, and to cite it in my own future works. Thank you to the authors, and well done, for this care in preparation!

This paper tackles the important and fascinating question of how molecular evolution would vary were amino acid "meanings" distributed to genetic code words (codons) differently than the pattern found within the standard genetic code. The standard genetic code had become established as the single, dominant pattern by which genes are translated into proteins by the time of LUCA. Molecular evolution for more or less all life on Earth has proceeded under the influence of this pattern for 3.5 billion years. The influence of the pattern is evident within all statistical summaries of accepted mutations, since the inception of the PAM matrix, and yet a wealth of evidence indicates that the pattern was itself an outcome of evolution, not a biophysical constraint. Many different branches of biological science, from bioinformatics and synthetic biology to evolutionary theory and "origins", are therefore directly interested in the data presented by the question addressed by this manuscript. This important question is currently underdeveloped within the literature and the authors explain their significance well. Citations are thorough and provide an excellent guide for all readers to understand what has been done on all fronts (including the growing technological reality of engineered, artificial genetic codes). Indeed the general level and thoroughness of citations throughout this manuscript is one of its many strengths. Perhaps the one citation that I was surprised not to see in the introduction is to the deeply similar (though strictly analogous) question/investigation made by Stadlers, Fontana and Wagner for RNA: (J Theor Biol, (2001) Nov 21;213(2):241-74. doi: 10.1006/jtbi.2001.2423.

"The topology of the possible: formal spaces underlying patterns of evolutionary change")

Having established that the manuscript is well written, the question is of broad interest and that the significance of the work is well developed, the next observation I would offer is that the approach (methods) seem very sensible - impressively thoughtful in navigating the opportunities and limitations of current scientific capabilities, both computational and informational (i.e. the data that exists to serve an investigation of this question). I particularly like the use of the saturated sequence "maps" for two real proteins that are each involved in measurable binding as a proxy for fitness. I understand why it is only a few sites that can be considered and this is state of the art for all practical purposes. I do think this is a limitation worth pointing out more heavily in the discussion, if only to motivate future work on more powerful computing and/or richer phenotypic maps of the future.

Turning to results, my opinion is that the statistical approaches used to analyze results are equally well thought out - in particular, they are elegantly simple to derive insights appropriate to the limitations and assumptions of the study. Maybe that is a fancy way of saying I felt confident that I understood what was being measured, I could see its direct relevance to the conclusions being drawn and I was persuaded by the data. Well done! For what it is worth, for the exact reason stated in my previous sentence, the single metric that I found most compelling was the number of adaptive peaks - I am a little cynical about reading too much into more sophisticated metrics of smoothness: here, they are well presented; in other literature exploring evolvability, I have seen authors place too much faith in a highly sophisticated metric without remembering how dependent its meaning is upon their underlying assumptions. The current work skirts closest to this danger in Figure 5, but the thoughtful text in section 2.7 (and even lines 745-746) rescues readers from reading too much into premature over extrapolation of ideals for "the perfect code"!

Conclusions/Discussion

Sensible, well developed and do not extend beyond findings - in fact, and excellent job here to guide readers new to this thinking about what exactly has and has not been discovered here. 

Summary: A great paper, of wide interest - well done!

Other comments:

My only suggestion for substantive is very tentative, and really one for the editor and authors to discuss. This is a long paper, and I believe it could potentially work better as two back-to-back papers: Parts 1 and 2. The most obvious dividing line for me would seem to be "part 1: measuring the influence of the code on fitness landscapes" and "Part 2: simulating effect of the code on evolutionary pathways"

To be clear, there is nothing wrong with the paper as one big whole - My worry is that it might not get the readership it deserves (# readers, and depth of careful reading) if presented as one monolithic test. It is, to me, a "magnum opus" for this group's work over a long and careful period of development. I would humbly offer the opinion that both PLoS and the authors might benefit from finding a dividing line and breaking it into 2 chapters for the audience. 

Reviewer #2:

The paper by Rozhonova et al. addresses an interesting and important question: to what extent does robustness of the genetic code facilitate or hinder the production of adaptive variation. While this question has received prior treatment, it remains unanswered for a number of technical reasons. The current work addresses several of these issues by using data from combinatorically complete deep mutational scans to characterize protein function and explore how the shape of these empirically based landscapes change as the genetic code is computationally 'rewired'. Overall the work is well written, with sufficient detail to understand what was done, why it was done, and the conclusion that the authors' draw from the results. In general, I have very little to add that would likely improve the paper and I think it is a strong addition to the field. However, there are a few caveats to the work worth pointing out.

1. My largest concern with the work is the definition of robustness used and how this may influence the results. The definition of robustness relies on the long standing observation that some amino acids have more similar biochemical properties than others. Current, and previous work by others, uses this observation to code sets of amino acids as being equivalent and that mutations between those amino acids are robust, while changes between amino acids in different sets are not robust. I understand the use of this simplification, but I'm concerned about how it could influence the papers conclusions. In particular, it is well established biochemical similarities between some amino acids lead to greater exchangeabilities during evolution, but that the ability to exchange two amino acids at any particular site may be idiosyncratic. For example, I and V are routinely seen to flip back and forth in alignments and very often have similar effects on protein function in deep mutational scanning data sets. However, they aren't always exchangeable and this property depends on the particular protein and site examined. My concern is that robustness is measured based on global patterns, while evolvability is dependent on the specific data set analyzed and that this mismatch, and in particular the way in which the amino acids are divided into sets, may influence the conclusions. 

This concern is amplified by the authors' analysis of alternative ways to group sets of amino acids by hundreds of other properties. While very few of these sets resulted in the opposite conclusion that robustness generates more rugged landscapes, the majority resulted in no relationship between robustness and ruggedness. This is also not the exact same analysis as conducted earlier in the paper - it connects the different ways of sorting amino acids into groups to ruggedness, but not to evolvability itself. Given the weak correlations overall, how different ways of grouping the amino acids to establish robustness would affect evolvability isn't clear. Furthermore, the analysis as conducted was on the entire protein and not site specific, which again is the necessary scale on which the idiosyncrasies in amino acid exchangeability are observed.

One idea that could alleviate this concern would be to use the data itself to determine which amino acids are functionally similar for each protein at each particular site and then reanalyze the different genetic codes u

---

## [Decision Letter · Decision Letter 2]

15 Feb 2024

Dear Dr Payne,

Thank you for your patience while we considered your revised manuscript "Protein evolvability under rewired genetic codes" for publication as a Research Article at PLOS Biology. This revised version of your manuscript has been evaluated by the PLOS Biology editors, the Academic Editor and the original reviewers.

Based on the reviews, we are likely to accept this manuscript for publication, provided you satisfactorily address the remaining points raised by the reviewers, and the following data and other policy-related requests:

IMPORTANT - please attend to the following:

a) Please could you change your Title to something more explicit and declarative? We suggest something like "Robust genetic codes enhance protein evolvability"

b) Please address the remaining requests from the reviewers.

c) Please cite the location of the data clearly in all relevant main and supplementary Figure legends (by my understanding, that would be Figs 2ABC, 3ABC, 4ABCDE, 5ABCD, 6ABC, S2ABCDEF, S4ABCDEF, S5ABCDEF, S8, S9, S12AB, S13, S14, S16, S17, S18ABC, S19AB, S20ABCDEF, S21ABCDEF, S22, S23, S24, S25.), saying e.g. “The data and code required to generate this Figure can be found in https://doi.org/10.5281/zenodo.8120904” - this may seem repetitive, but it will make the Figs more standalone.

We expect to receive your revised manuscript within two weeks. 

*Published Peer Review History*

*Press*

Sincerely,

Roli Roberts

Roland Roberts, PhD

Senior Editor

rroberts@plos.org

PLOS Biology

CODE POLICY

Per journal policy, as the code that you have generated is important to support the conclusions of your manuscript, we require that you make it available without restrictions upon publication. Please ensure that the code is sufficiently well documented and reusable, and that your Data Statement in the Editorial Manager submission system accurately describes where your code can be found.

DATA NOT SHOWN?

REVIEWERS' COMMENTS:

Reviewer #1:

Apologies for a slow response here as commitments became rather overwhelming in the past couple of weeks. 

I fully support publication of this paper - I think the points raised by reviewers are valid, especially as summarized by the editor. I think that the authors have addressed these points, and that the paper is publishable. I do have very minor comments (really the suggestion to add ~2 important sentences, that continue/strengthen the response to these comments. 

Editor comment 1: "Shorten the manuscript..."

Reviewer comment from me: The significant reduction definitely makes a more readable paper to this reader: thank you! I do regret that lots of important information is now in the supplementary data - but that is exactly what was asked for. So in that context, I probably haven't truly let go of my original impression that there were two papers here: one (lower impact factor) that sets up all the assumptions and methods, with checks and validations; then a second paper focusing on the story (results) presented here. I say that because the remainder of my comments below really just ask the authors if they could be careful to highlight specific elements of this important supplementary information for careful readers - this background material is I think crucial to a healthy and accurate reading of the work as a whole. 

Editor comment 2: "Address the impact of the method used to calculate robustness and clarify some of the methods that are missing or unclear (as suggested by reviewer 2). Which physicochemical properties were used, and which ones were most important?"

Reviewer comment from me: Again, I am 100% in support of the editor's redux of the reviewer's point - very well spotted by the reviewer, and the more I dug into this point the more interested I was by the (apparent) answers. I now perceive that within the discussion, under the paragraph "several caveats…", it merits a sentence that highlights the nature of the simplification and where it can be read about in much greater detail in supplementary material. If I understand correctly, then robustness of the code, measured around the concept of "fraction of point mutations which do NOT cause a change in amino acid meaning" also simplifies the 20 amino acid meanings into groups which are not truly synonymous (from the perspecitve of traditional molecular evolution) - this simpliification scheme is inherited from Pines et al. [22] (Supp. Fig. 725 S3), right? Fascinating, but definitely something a naive reader needs to bear in mind thinking about the results here.

Editor comment (3): "Clearly discuss the very weak correlations and different outcomes for the different proteins, and resulting biological implications, as indicated by reviewer 3. I think these issues do reduce the novelty and impact of the results; at the very least they should be clearly discussed. The low dimensionality of the dataset (reviewer 4) is similarly an important concern that should be discussed. 

R1: Once again I do see the editor/reviewer's point and see/appreciate the steps taken by the authors. Once again, however, I would encourage the authors to add a specific sentence to the Discussion (probably paragraph beginning "There are several caveats to the results presented here…"?) reminding readers that results vary for the 4 proteins, and directing them specifically to the supplementary information where this can be considered in details (e.g. S2 Analysis of physicochemical properties from the Aaindex?)

Reviewer #2:

I'm satisfied with the author's responses to the reviews. The paper is a great addition to the field.

Reviewer #3:

[identifies himself as Yitzhak Pilpel]

The authors have revised the paper according to comments made by the 3 referees. Obviously the other referees are more positive than myself. 

I do appreciate very much their thought and response to my comments. Adding three more proteins is a good addition to their test set. I remain under impressed due to the two main caveats I saw in the first submission: still very few proteins examined. I still believe that they could have tested proteins whose mapping was incomplete (that don't meet the "20^L" criterion), that would have been a methodological achievement by itself, and could allow them to examine many more proteins. It is still the case that the results are very very moderate in magnitude, and still a very small % of the variance is explained. Their response to my question of generality "As stated at the end of Section 2.2.1, this illustrates that the influence of a genetic code on protein evolvability is protein-specific" confuses me very much. The whole discussion of the code supporting evolvability could imply some generality, but the end result is lack thereof , with only 6 proteins examined. 

But I'm not against publication. The paper and study, as said originally, is very intelligently done, it will contribute to the study of the code and will surely be followed up by others.

Reviewer #4:

The authors have addressed my concerns. I am happy for this paper to be published in PLOS Biology.

---

## [Editor Report · Decision Letter 3]

19 Mar 2024

Dear Dr Payne,

Thank you for the submission of your revised Research Article "Robust genetic codes enhance protein evolvability" for publication in PLOS Biology. On behalf of my colleagues and the Academic Editor, Deepa Agashe, I'm pleased to say that we can in principle accept your manuscript for publication, provided you address any remaining formatting and reporting issues. These will be detailed in an email you should receive within 2-3 business days from our colleagues in the journal operations team; no action is required from you until then. Please note that we will not be able to formally accept your manuscript and schedule it for publication until you have completed any requested changes.

Sincerely, 

Roli Roberts

Senior Editor

PLOS Biology

rroberts@plos.org